# The impact of functional medicine on patient-reported outcomes in inflammatory arthritis: A retrospective study

Nicole Droz[1], Patrick Hanaway[2], Mark Hyman[2], Yuxuan Jin[3], Michelle Beidelschies[2], M. Elaine Husni[1] *

1 Orthopaedic and Rheumatologic Institute, Cleveland Clinic, Cleveland, OH, United States of America, 2 Center for Functional Medicine, Cleveland Clinic, Cleveland, OH, United States of America, 3 Quantitative Health Sciences, Cleveland Clinic, Cleveland, OH, United States of America

* HusniE@ccf.org

## Abstract

### Background

Despite treatment advances for inflammatory arthritis, a significant amount of patients fail to achieve remission. Other modifiable factors such as diet, physical activity and environmental exposures may be an important area of focus to help patients achieve disease remission and greater overall health. Functional medicine focuses on these lifestyle factors and may be an important adjunctive therapy. In this study, we examined the impact of functional medicine on patient-reported outcomes in patients with inflammatory arthritis.

### Materials and methods

In this 12-week, retrospective study, patients with confirmed diagnoses of rheumatoid arthritis (RA) or psoriatic arthritis (PsA) were treated according to guidelines from the American College of Rheumatology for RA or PSA respectively. Those in the functional medicine group underwent a functional medicine program adjunctive to the standard of care. Patient reported outcomes, such as PROMIS (Patient Reported Outcomes Measurement Information System) global physical health, mental health and pain scores were collected at baseline and 12 weeks. Multivariable statistical modeling was used to identify the impact of functional medicine on patient-reported outcomes.

### Results

318 patients were screened and 54 patients (mean age 52.9±11.3 years, females 74 (67.9%)), were included. Baseline characteristics were similar in both patient groups with the exception of PROMIS global physical health and pain (PROMIS global physical health score 43·2 ± 6·6 and 39·7 ± 8·7 and pain scores of 3·5 ± 1·9 and 5·2 ± 2·7 in the functional medicine group vs. standard of care group respectively). Using multivariable model to account for these differences, patients in the functional medicine group had a statistically significant reduction in pain (0.92, p-value = 0.007) and change in PROMIS physical health score (2·84, p-value = 0.001) as compared to the standard of care. Changes in PROMIS

**Data Availability Statement:** Data used for the generation of this study includes human research participant data that are sensitive and cannot be publicly shared due to legal and ethical restrictions

by the Cleveland Clinic regulatory bodies including the institutional review Board and legal counsel. Some variables collected were HIPAA protected health information and legally cannot be publicly shared. Since these variables were critical to the analyses, a partial dataset (everything except them) is not fruitful either because it will not help in efforts of academic advancement, such as model validation or application. We will make our data sets available upon request, under appropriate data use agreements with the specific parties interested in academic collaboration. Requests for data access can be made to Sandra Hodnick (HODNICS@ccf.org).

**Funding:** The author(s) received no specific funding for this work.

**Competing interests:** Nicole Droz, MD- none Yuxuan Jin- none Mark Hyman, MD- * Ownership of stocks or shares: Mutual funds and investments that may include medically or food related companies managed by investment broker, multiple private companies including Thrive Market, Hu Kitchen, Walden Meats, WTR MLN WTR, Parsley Health, Bulletproof, Good Money * Paid employment or consultancy: Cleveland Clinic and owner of The UltraWellness Center, Hyman Digital, Hyman Enterprises (published 16 books on health and nutrition), Vitamin Portfolio, Farmacy, The Doctor's Farmacy podcast * Board membership: Institute for Functional Medicine, Environmental Working Group, Center for Mind Body Medicine Michelle Beidelschies, PhD: reported receiving personal fees from Cleveland HeartLab, Inc. outside the submitted work. In addition, Dr. Beidelschies had a patent (No. 20110269150) issued for compositions and methods for predicting cardiovascular events. M. Elaine Husni, MD MPH: Dr. Husni is a consultant (with honoraria) from AbbVie, Janssen, Sanofi Genzyme/Regeneron, UCB, Novartis, and Lilly (less than $10,000 each) and is a coinventor on a psoriatic arthritis questionnaire PASE (Psoriatic Arthritis Screening Evaluation) which is copyrighted, for which she receives royalties. Patrick Hanaway, MD: Dr Hanaway reported serving as paid educational consultant and teaching for the Institute for Functional Medicine. This does not alter our adherence to PLOS ONE policies on sharing data and materials.

global mental health scores were also significant and were dependent on age and were greatest in those older than 55.

## Limitations

Retrospective design, baseline difference in patient reported outcomes.

## Conclusions

Functional medicine may have an important role as adjunctive therapy to improve patients' pain, physical and mental health in those who do not see improvement with conventional therapy alone.

## Introduction

Rheumatoid and psoriatic arthritis are chronic, multisystem, inflammatory disorders associated with significant morbidity. In the United States, arthritis of all causes remains the leading cause of disability accounting for more than 35% of total disability [1]. The treatment landscape has changed dramatically over the past decade. The addition of biologic and non-biologic DMARDs (disease modifying anti rheumatic drugs) for the treatment of inflammatory arthritis has greatly improved patient outcomes; however, patients frequently do not achieve clinical remission of their disease [2]. There is still no cure for RA or PsA and these chronic diseases can take a toll over the patient's lifetime.

In addition to the traditional physician-assessed disease activity, patient reported outcomes can be critical to assess a patient's overall health. A patient global assessment (PtGA) of disease is often a limiting factor in achieving ACR/EULAR remission [2, 3]. Disease activity and functional status assessments currently utilized in patient care fail to address all treatment outcomes that are important to patients (such as fatigue, mental health, pain, and quality of life) [4]. These outcomes drive PtGA and represent important areas of focus to achieve overall remission and well-being in a greater proportion of patients [5].

The pathogenesis of inflammatory arthritis is complex and influenced by genetic and environmental factors. In patients with inflammatory arthritis, the immune response is dysregulated, leading to a state of chronic inflammation. Diet, physical activity, microbiome dysbiosis, and environmental exposures are just a few factors that have been implicated in driving chronic systemic inflammation, contributing to disease activity in inflammatory arthritis [6]. Since there are many potentially modifiable lifestyle factors that may affect patients with RA and PsA, improving health behaviors and lifestyle interventions could have benefits beyond traditional DMARD therapies.

The functional medicine model of care expands upon the conventional medicine model of care by addressing underlying causes, symptoms and functional imbalances associated with various biological functions. To remedy imbalances, functional medicine uses food as a first line therapy to prevent, treat and reverse chronic disease [7]. It provides an individualized, patient-centered model of care that organizes a patient's signs and symptoms along with their lifestyle, environmental, and genetic factors to identify underlying drivers of their chronic disease [7]. Specifically for patients with inflammatory arthritis, the initial approach includes a detailed medical history including birth history, antibiotic use, environmental exposures, digestive history, infection history, and initial testing derived from the medical history including nutrient status, gluten antibodies, microbiome analysis, toxin exposure, and hidden infections as indicated by the history

and clinical presentation. The practitioners treat patients using nutrition as an adjunctive to first-line therapy focusing on an anti-inflammatory, low glycemic index, nutrient-dense food plan that encourages the consumption of fresh, bright-colored vegetables, low glycemic index fruits, and non-processed foods. Dietary supplements are commonly used to balance nutritional deficiencies, as well as utilizing plant-based anti-oxidants, anti-inflammatory herbs, and prebiotics/ probiotics to modify the gut microbiome. Functional medicine also provides patients with a multi-disciplinary team of caregivers including a practitioner, registered dietitian and health coach to help them with lifestyle modification and behavior change, in order to reach their health goals.

Because the functional medicine model focuses on modifying multiple elements that drive chronic inflammation, we evaluated whether a functional medicine approach as an adjunct to standard therapy may improve symptoms of pain, psychological distress, and decreased functional capacity in patients with PsA and RA, outcomes critical to the PtGA and achieving disease remission.

## Materials and methods

### Study design and patients

In this retrospective, single center study, we evaluated the change in Patient Reported Outcomes Measurement Information System (PROMIS®) global physical health, PROMIS global mental health, and pain scores in patients who underwent a functional medicine program (n = 54) adjunctive to the standard of care as compared to patients who received standard of care alone (n = 55). The study protocol was reviewed and approved by the Institutional Review Board at the Cleveland Clinic Foundation (IRB#17–237). Patients aged 18 and older who were seen as a new patient in the Cleveland Clinic Center for Functional Medicine between August 2016 and January 2018 were screened for inclusion into the study by diagnosis code for RA (ICD 9: 714·0; ICD 10: M06·9) or PsA(ICD 9: 696·0; ICD10: L40·50). Patients were eligible for inclusion if they had documentation by a board certified rheumatologist at our institution confirming their diagnosis of RA or PsA and were on stable doses of prednisone and DMARD therapy throughout the study period. Patients were compared to controls seen only in the Rheumatology Department during the same time period. Patients in the standard of care group were matched to the functional medicine group based on similarity in baseline characteristics, such as age, sex, smoking status, and disease seropositivity, if applicable.

All patients received standard of care for RA and PsA and followed treatment regimens outlined by their treating rheumatologist. Treatments followed standard guidelines as defined by the American College of Rheumatology [8, 9].

The functional medicine diagnostic and treatment approach to RA and PsA focused on improving nutrition and digestive function, removing damaging environmental exposures, and providing longitudinal coaching support by way of a multi-disciplinary team of caregivers. All patients were instructed to follow an anti-inflammatory elimination diet (specifically no processed foods, sugar, gluten and dairy) based on a majority of whole, low glycemic index foods, good fats (MUFA's and omega 3 fats), fiber and phytonutrients. Nutrient levels were optimized and guided by laboratory results such as vitamin D or omega 3 fatty acids. If patients had altered intestinal permeability, assessed through stool tests or antibodies to lipopolysaccharide. In addition, patients with elevated levels of heavy metals such as mercury or lead (measured in whole blood or urine) would be treated to remove heavy metals. Patients saw a physician or advanced practitioner, registered dietitian and health coach as part of the initial visit and afterwards for ongoing support and to make adjustments to the program based on patient response and laboratory evaluations.

PROMIS global physical and mental health, and pain scores were collected at baseline and after 12 weeks of enrollment using CAT (computer assisted technology) questionnaires.

### Data collection

Data was collected by retrospective chart review and was stored in a RedCap database.

### Patient-reported outcomes

The outcomes of the study were 12 weeks to baseline changes in pain and PROMIS global physical and mental health scores. PROMIS is a set of validated patient-centered measures, developed by the National Institute of Health, that evaluate and monitor physical, mental and social health. This patient self-administered questionnaire was created to be relevant across all conditions for assessment of functionality, symptoms, behaviors and feelings [10]. In particular, this study utilized the PROMIS Scale v1·2- Global Health measure at baseline and at 12 weeks. PROMIS is a precise and reliable way to measure domains of physical, mental and social health that are critical to PtGA. Pain scores were collected using visual analog scale ranging from 0–10 at baseline and 12 weeks.

### Statistical analysis

Sample size was calculated using a two-sample t-test with a 5-point differential in mean PROMIS score changes, and a standard deviation of 5.15 [11]. This provided 80% power at the 5% significant level. The sample size required to achieve adequate power for PROMIS outcomes was 36 patients total, assuming 18 patients in each group. To account for a 10% attrition rate, it was recommended that at least 20 patients be assigned to each group (40 patients, total).

Summary statistics were calculated using means and standard deviations for continuous variables, such as age and patient reported outcomes. The normality of the continuous variables was tested using Shapiro-Wilk tests. Two-sample t-tests were used to compare the differences in continuous variables that meet the normality assumption, such as Global Physical and Mental Health T-scores at 12 weeks and the 12-week to baseline change in Global Physical Health T-score. Categorical variables, such as sex, smoking status anti-cyclic citrullinated peptide, rheumatoid factor were summarized using frequencies and percentages and were compared using Pearson's chi-square tests. To obtain the effect sizes, Cohen's ds were provided for patient reported outcomes at 12 weeks and the changes in scores. Three separate multiple linear regressions were used to model the association between 12 weeks to baseline changes in patient reported outcomes (PROMIS physical health, mental health and pain scores) and treatment group (functional medicine vs. standard of care). The covariates included in the full models were age, sex, smoking status, anti-cyclic citrullinated peptide, rheumatoid factor and corresponding baseline score. Moreover, interaction term between treatment group and each of these covariates were also included. Before building the models, multicollinearity among predictors was checked using variance inflation factors (VIF) and condition indices. VIF greater than 10 and condition indices greater than 30 were used to identify strong multicollinearity. No multicollinearity was found. Backward elimination (with a threshold of $p < 0.05$) was used to identify important variables in the models and results of the reduced models were shown. Data management, descriptive statistics and multivariable modeling were performed using SAS software (version 9·4; Cary, NC). All tests were two-sided, with an alpha-level of 0.05.

## Results

### Baseline characteristics of study groups

318 patients seen in the functional medicine department were initially identified as having RA or PsA by ICD 9/10 code. Of these patients, 54 had documentation of their diagnosis by a board certified rheumatologist and had complete data over the study time frame and were included into the

**Table 1. Descriptive statistics on patient reported outcomes at baseline, demographic and clinical characteristics.**

| Factor | Overall (N = 109) | | Functional Medicine (N = 54) | | Standard of Care (N = 55) | | |
|---|---|---|---|---|---|---|---|
| | N | Statistics | n | Statistics | n | Statistics | p-value |
| Global physical health T-score baseline | 109 | 41.4±7.9 | 54 | 43.2±6.6 | 55 | 39.7±8.7 | *0.022*[a] |
| Global mental health T-score baseline | 109 | 45.6±8.3 | 54 | 45.8±8.4 | 55 | 45.3±8.3 | 0.73[a] |
| Pain score baseline | 108 | 4.4±2.5 | 54 | 3.5±1.9 | 54 | 5.2±2.7 | *<0.001*[a] |
| RAPID3 baseline | 51 | 3.9±2.0 | 6 | 3.2±1.02 | 45 | 4.0±2.1 | 0.36[a] |
| Age | 109 | 52.9±11.3 | 54 | 51.0±10.6 | 55 | 54.8±11.8 | 0.078[a] |
| Sex | 109 | | 54 | | 55 | | 0.075[c] |
| male | | 35(32.1) | | 13(24.1) | | 22(40.0) | |
| female | | 74(67.9) | | 41(75.9) | | 33(60.0) | |
| Smoking | 108 | | 53 | | 55 | | 0.16[c] |
| Never | | 62(57.4) | | 34(64.2) | | 28(50.9) | |
| Former or Current | | 46(42.6) | | 19(35.8) | | 27(49.1) | |
| CCP | 109 | | 54 | | 55 | | 0.67[c] |
| Positive | | 43(39.4) | | 23(42.6) | | 20(36.4) | |
| Negative | | 40(36.7) | | 20(37.0) | | 20(36.4) | |
| Unknown | | 26(23.9) | | 11(20.4) | | 15(27.3) | |
| RF | 109 | | 54 | | 55 | | 0.12[c] |
| Positive | | 37(33.9) | | 22(40.7) | | 15(27.3) | |
| Negative | | 50(45.9) | | 25(46.3) | | 25(45.5) | |
| Unknown | | 22(20.2) | | 7(13.0) | | 15(27.3) | |
| Treatment at baseline | 108 | | 53 | | 55 | | *<0.001*[C] |
| Conventional DMARD therapy | | 29(26.9) | | 12(22.6) | | 17(30.9) | |
| Biologic DMARD therapy | | 66(61.1) | | 28(52.8) | | 38(69.1) | |
| None | | 13(12.0) | | 13(24.5) | | 0(0.0) | |

Statistics presented as Mean ± SD or N (column %).

p-values: a = ANOVA, c = Pearson's chi-square test.

CCP = anti-cyclic citrullinated peptide; RF = rheumatoid factor; DMARD = Disease modifying anti-rheumatic drug

study. The main reasons for exclusion from the study were: incomplete data, inadequate follow up within the first 12 weeks after starting the functional medicine program, inadequate documentation of rheumatologic diagnosis or confounding by lack of stable doses of medications to treat inflammatory arthritis. In the standard of care group, 54 patients seen in the rheumatology department alone without any functional medicine intervention were identified for inclusion into the study. Baseline characteristics were similar between both groups with the exception of PROMIS global physical health and pain scores which were significantly different in the functional medicine group as compared to the standard of care group (PROMIS global physical health score 43·2 ± 6·6 in the functional medicine group as compared to 39·7 ± 8·7 in the standard of care group and pain scores of 3·5 ± 1·9 and 5·2 ± 2·7 respectively) (Table 1). There was also a statistically significant difference among treatment regimens. This was likely due to a high number of patients in the functional medicine group who were risk averse to using chronic disease modifying therapy. Interestingly disease activity scores as measured by RAPID3, a validated disease assessment instrument, were similar despite these treatment differences.

## Univariate analyses evaluating the effects of functional medicine on PROMIS physical, mental and pain scores at 12 weeks

On univariate analysis, 12 week PROMIS global physical health T scores (45·4 ±6·5 and 40·2 ± 7·1) and pain scores (3·1 ± 1·8 and 5·0 ± 2·4) remained statistically significant between the functional medicine and standard of care groups respectively, however, there was no

**Table 2. Descriptive statistics on patient reported outcomes at 12 weeks, and change (12 weeks–baseline) in scores.**

| | Overall (N = 109) | | Functional Medicine (N = 54) | | Standard of Care (N = 55) | | | |
|---|---|---|---|---|---|---|---|---|
| Factor | N | Statistics | N | Statistics | N | Statistics | Cohen's d | p-value |
| PROMIS global physical health T score at 12 weeks | 109 | 42.8±7.3 | 54 | 45.4±6.5 | 55 | 40.2±7.1 | 0.76 | <*0.001*[a1] |
| PROMIS global mental health T score 12 weeks | 109 | 46.5±8.5 | 54 | 46.8±8.9 | 55 | 46.1±8.2 | 0.08 | 0.70[a1] |
| Pain 12 weeks | 108 | 4.1±2.3 | 54 | 3.1±1.8 | 54 | 5.0±2.4 | -0.90 | <*0.001*[a2] |
| RAPID 3 12 weeks | 52 | 3.8±2.0 | 6 | 2.1±0.78 | 46 | 4.0±2.0 | -0.99 | *0.021*[a2] |
| PROMIS global physical health score change | 109 | 1.3±5.0 | 54 | 2.2±4.9 | 55 | 0.48±5.0 | 0.35 | 0.068[a1] |
| PROMIS global mental health score change | 109 | 0.90±4.7 | 54 | 0.94±4.8 | 55 | 0.86±4.7 | 0.02 | 0.93[a1] |
| Pain score change | 108 | -0.31±1.9 | 54 | -0.41±1.6 | 54 | -0.20±2.1 | -0.11 | 0.57[a1] |

Statistics presented as Mean ± SD.

p-values: a1 = t-test, a2 = Satterthwaite t-test.

statistically significant change when comparing groups to each other. In terms of the effect sizes (Cohen's d), at univariate level, 12-week pain scores and RAPID 3 had biggest effect sizes of around 0.9, suggesting that the mean differences of these two measures equal to around 0.9 of standard deviation (Table 2).

## Multivariate analyses evaluating the effects of functional medicine on PROMIS physical, mental and pain scores at 12 weeks

Table 3 showed that average change of physical health score in functional medicine group was 2.84 point greater compared to standard of care group (p = 0.001), after controlling for baseline physical health score.

The average change of pain score in functional medicine groups was 0.92 point less compared to standard of care group (p = 0.007), after controlling for baseline pain score. Therefore, physical health and pain scores both showed significant improvement in the functional medicine group as compared to the standard of care group. After adjusting for the difference in baseline mental health scores, there was also a significant improvement in mental health in the

**Table 3. Results for multivariable models.**

| Term | Coefficient | 95% CI | p-value |
|---|---|---|---|
| *Model 1: Change in PROMIS Global Physical Health score* | | | |
| Intercept | 13.02 | (8.58, 17.46) | <*0.001* |
| Treatment group: Functional Medicine | 2.84 | (1.15, 4.53) | *0.001* |
| Baseline PROMIS Global Physical Health Score | -0.32 | (-0.42, -0.21) | <*0.001* |
| *Model 2: Change in Pain Score* | | | |
| Intercept | 1.89 | (1.07, 2.72) | <*0.001* |
| Treatment group: Functional Medicine | -0.92 | (-1.58, -0.26) | *0.007* |
| Baseline Pain Score | -0.4 | (-0.53, -0.27) | <*0.001* |
| *Model 3: Change in PROMIS Global Mental Health Score* | | | |
| Intercept | 15.42 | (8.30, 22.55) | <*0.001* |
| Treatment group: Functional Medicine | -12.97 | (-21.04, -4.89) | *0.002* |
| Age | -0.15 | (-0.25, -0.05) | *0.003* |
| Baseline PROMIS Global Mental Health Score | -0.14 | (-0.24, -0.04) | *0.006* |
| Age*Treatment group: Functional Medicine | 0.24 | (0.09, 0.39) | *0.002* |

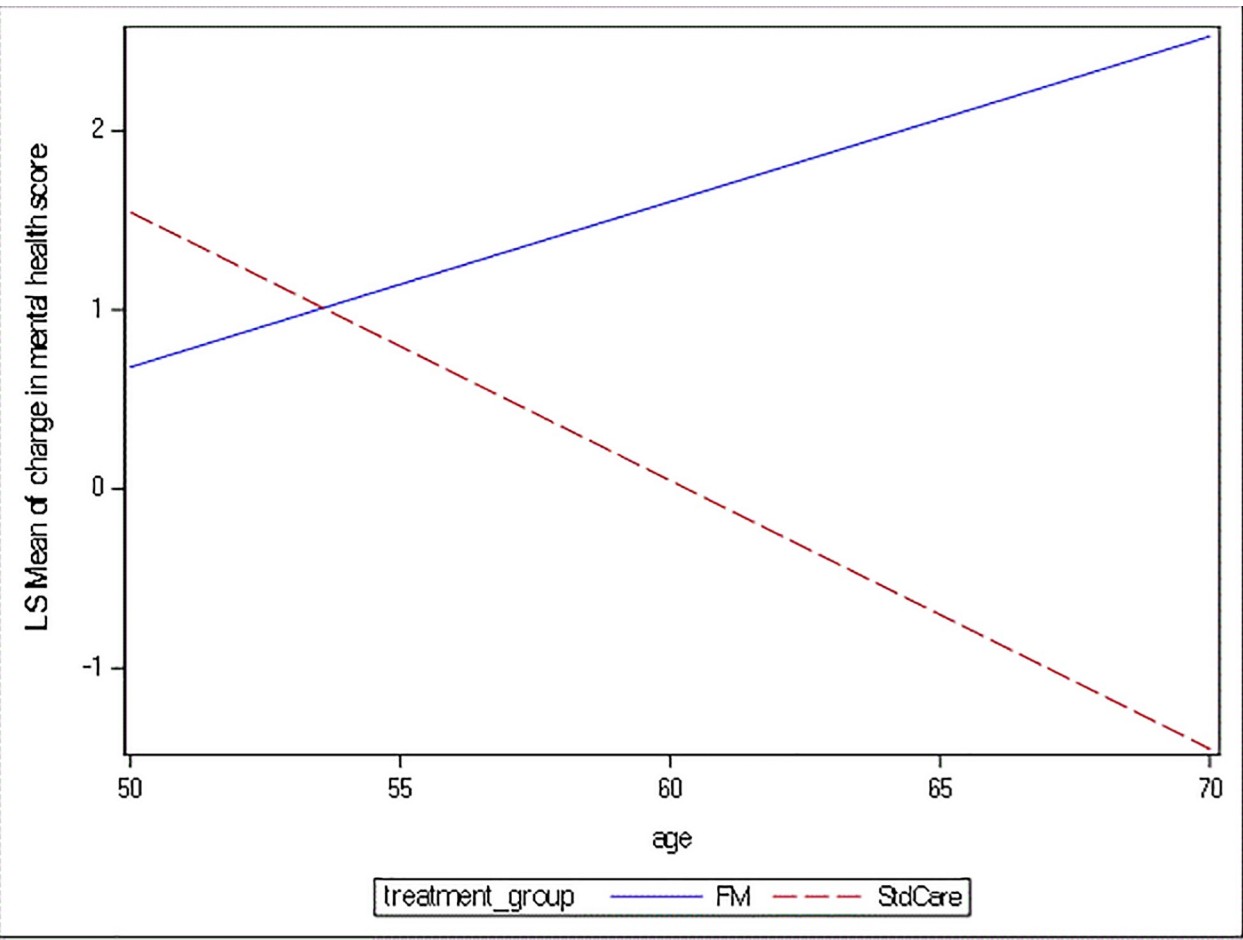

**Fig 1. Mean change in mental health score vs. age.**

functional medicine group as compared to standard of care. The significant age * treatment group (p = 0.002) interaction term suggested that mental health was dependent on age and the impact of an adjunctive functional medicine treatment was seen to have the greatest benefit in patients aged 55 and older (Fig 1).

This figure displays the significant interaction between age and treatment group for change in mental health score. Y-axis is the least square mean estimate of change in mental health score. The difference in slope of the two lines suggests that the effect of functional medicine on mental health depends on age. When age progresses, the role of functional medicine becomes obvious, as suggested by the increase of change in mental health score.

## Discussion

In this retrospective study, the adjunctive treatment with a 12-week functional medicine program improved PROMIS physical and mental health scores as well as pain scores as compared to the standard of care. These results are important new knowledge for healthcare professionals when discussing adjunctive care related to nutrition and lifestyle habits in RA and PsA patients. Since prior research has been largely limited to both anecdotal and observational analyses for these alternative care models, these results support the use of adjunctive care with a functional medicine approach to improve patient reported outcomes in RA and PsA patients.

Due to the baseline characteristic differences and wide distribution of responses, a multivariable analysis was performed and demonstrated a statistically significant improvement in pain scores of -0.92 in the functional medicine group as compared to the standard of care group. Previous observational studies have reported the minimal clinically significant difference in pain scores in patients with rheumatoid arthritis as a reduction of pain on a 10 point VAS scale of -0.5 to -1.1 [12]. Therefore, although the reductions in pain score in the functional medicine group were small, they were both clinically and statistically significant.

Similarly, PROMIS physical health T scores were greater by 2.84 points in the functional medicine group as compared to standard of care when controlling for baseline differences. Previous studies have suggested that the minimal clinically significant difference in PROMIS physical health scores range from 1.9 to 2.2 indicating this was also both statistically and clinically significant [13].

Interestingly, improvements in PROMIS mental health score were dependent on age. Compared to those less than 55 years of age, patients older than 55 showed an improvement in PROMIS mental health scores with adjunctive functional medicine. This may have important clinical implications and suggests that older patients may have an added benefit from the addition of a functional medicine program compared to younger patients.

Mental health-related illness is common in the elderly and may arise secondary to associated medical comorbidities, poor socioeconomic status, disability and/or social isolation [14]. Additionally, those with a diagnosis of rheumatoid arthritis are two to four times more likely to experience major depressive disorder than the general population [15]. Recent evidence suggest that improved nutrition may in fact help alleviate depression. Numerous studies have demonstrated that higher intakes of fruit, vegetables, and fish may be associated with reduced depression risk [16, 17]. In addition, a recent randomized controlled trial in a middle-aged population (early 40s) demonstrated a significant reduction in depression symptoms following a dietary intervention that included personalized support from a dietitian and the consumption of a modified Mediterranean diet [18]. Thus, dietary and lifestyle improvements, such as those used in functional medicine, may have a role in the improvement of mental health scores in elderly patients.

Our study had several limitations. First, was the retrospective nature of our study. Because patients were not randomized, the patients who participated in the functional medicine program may have an inherent bias towards not using chronic disease modifying therapy. This was illustrated by the differences in baseline treatment regimens. More patients in the functional medicine group were on no RA or PsA disease modifying therapy as compared to patients in the standard of care group. RAPID3 scores were similar between groups indicating that this difference in treatment was not reflective of differences in disease activity. Without prospectively randomizing patients, we were unable to achieve homogeneity regarding treatment or certain lifestyle behaviors between groups. For instance, a higher number of patients in the standard of care group were smokers as compared to patients in the functional medicine group. Although this was not statistically significant, it is well known that cigarette smoking increases rheumatoid arthritis disease severity [19]. It is possible that the statistically significant differences in PROMIS physical health and pain scores favoring the functional medicine group is reflective of milder disease or shorter duration of disease as opposed to those patients receiving standard of care in rheumatology alone. However, using a multivariate analysis, we were able to account for these differences seen between groups and were able to demonstrate a statistically significant improvement in all primary outcomes in patients treated with an adjunctive functional medicine program.

This study represents the first report of the positive impact of functional medicine on patient reported outcomes in inflammatory arthritis patients. These findings indicate that

functional medicine may have an important role as an adjunctive therapy to address patient's physical and mental health, as well as pain, in those who have not seen improvement with standard of care alone. Due to the individualized nature of patient care plans in the Center for Functional Medicine, it is difficult to speculate which particular component of functional medicine treatment had the greatest impact on patient reported outcomes, or whether it was due to other personalized treatment variables. Further research is needed to elucidate which aspect(s) of functional medicine is responsible for the positive results seen in our study.

## Acknowledgments

We would like to acknowledge the Center for Functional Medicine at Cleveland Clinic Foundation for their support with this project. We would specifically like to recognize Marilyn Alejandro-Rodriguez for her efforts.

## Author Contributions

**Conceptualization:** Nicole Droz, Patrick Hanaway, Mark Hyman, Yuxuan Jin, M. Elaine Husni.

**Data curation:** Nicole Droz, M. Elaine Husni.

**Formal analysis:** Nicole Droz, Yuxuan Jin, M. Elaine Husni.

**Methodology:** Nicole Droz, Patrick Hanaway, Yuxuan Jin, M. Elaine Husni.

**Supervision:** Patrick Hanaway, M. Elaine Husni.

**Writing – original draft:** Nicole Droz, Patrick Hanaway, Mark Hyman, Yuxuan Jin, Michelle Beidelschies, M. Elaine Husni.

**Writing – review & editing:** Nicole Droz, Patrick Hanaway, Mark Hyman, Yuxuan Jin, Michelle Beidelschies, M. Elaine Husni.

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
