## [Decision Letter · Decision Letter 0]

6 Jul 2020

PONE-D-20-14349

The Impact of Functional Medicine on Patient-Reported Outcomes in Inflammatory Arthritis: A Retrospective Study

PLOS ONE

Dear Dr. Nicole Droz

Thank you for submitting your manuscript to PLOS ONE. After careful consideration, we feel that it has merit but does not fully meet PLOS ONE’s publication criteria as it currently stands. Therefore, we invite you to submit a revised version of the manuscript that addresses the points raised during the review process.

This a relatively small study, The research hypothesis, sample size calculation, power analyses and the methodology used in the study need to be re-evaluated.

Please address all of the attached comments. 

We look forward to receiving your revised manuscript.

Kind regards,

Mahmoud Abu-Shakra, MD

Academic Editor

PLOS ONE

Journal Requirements:

'I have read the journal's policy and the authors of this manuscript have the following competing interests:

Nicole Droz, MD- none

Yuxuan Jin- none

Mark Hyman, MD- * Ownership of stocks or shares:  Mutual funds and investments that may include medically or food related companies managed by investment broker,, multiple private companies including Thrive Market, Hu Kitchen, Walden Meats,  WTR MLN WTR, Parsley Health, Bulletproof, Good Money

* Paid employment or consultancy: Cleveland Clinic and owner of The UltraWellness Center, Hyman Digital, Hyman Enterprises (published 16 books on health and nutrition), Vitamin Portfolio, Farmacy, The Doctor’s Farmacy podcast

* Board membership: Institute for Functional Medicine, Environmental Working Group, Center for Mind Body Medicine

Michelle Beidelschies, PhD: reported receiving personal fees from Cleveland HeartLab, Inc. outside the submitted work. In addition, Dr. Beidelschies had a patent (No. 20110269150) issued.

Elaine Husni, MD: Dr. Husni is a consultant (with honoraria) from AbbVie, Janssen, Sanofi Genzyme/Regeneron, UCB, Novartis, and Lilly (less than $10,000 each) and is a coinventor on a patent for a psoriatic arthritis questionnaire PASE (Psoriatic Arthritis Screening Evaluation), for which she receives royalties.

Patrick Hanaway, MD: Dr Hanaway reported serving as paid educational consultant and teaching for the Institute for Functional Medicine.'

We note that you have a patent relating to material pertinent to this article.

Please provide an amended statement of Competing Interests to declare this patent (with details including name and number), along with any other relevant declarations relating to employment, consultancy, patents, products in development or modified products etc.

Please confirm that this does not alter your adherence to all PLOS ONE policies on sharing data and materials, as detailed online in our guide for authors http://journals.plos.org/plosone/s/competing-interests by including the following statement: "This does not alter our adherence to  PLOS ONE policies on sharing data and materials.” If there are restrictions on sharing of data and/or materials, please state these. Please note that we cannot proceed with consideration of your article until this information has been declared.

4. Please justify the sample size included in this study, i.e. with reference to any sample size calculations performed or by citing previous studies.

Please refer to our statistical reporting guidelines for assistance (https://journals.plos.org/plosone/s/submission-guidelines.#loc-statistical-reporting).

Additional Editor Comments:

Add the limitations of study in the abstract and discussion.

Calculate sample size and perform power analyses.

The paper needs to be reviewed by a statistician to address whether the methodology used can address the research objectives

Reviewers' comments:

Reviewer's Responses to Questions

**Comments to the Author**

1. Is the manuscript technically sound, and do the data support the conclusions?

Reviewer #1: Yes

Reviewer #2: Partly

Reviewer #3: Yes

2. Has the statistical analysis been performed appropriately and rigorously? 

Reviewer #1: I Don't Know

Reviewer #2: No

Reviewer #3: Yes

3. Have the authors made all data underlying the findings in their manuscript fully available?

Reviewer #1: Yes

Reviewer #2: Yes

Reviewer #3: Yes

4. Is the manuscript presented in an intelligible fashion and written in standard English?

Reviewer #1: Yes

Reviewer #2: Yes

Reviewer #3: Yes

5. Review Comments to the Author

Reviewer #1: Grammatical errors (per Line number):

1. L30: “Despite, treatment” –remove coma

2. L30:“arthritis ,” -- remove empty space

3. L34: “patient reported” --hyphenate

4. L39: “Multivariable”—should probably be the more common MULTIVARIATE

5. L40: be consistent with either “patients reported” or patient-reported

6. L49: “patient’s” – should be patient or patients’

7. L61: “outcomes, however” –use semicolon or start new sentence

8. L65: “physician assessed” --hyphenate

9. L115: “regiment” is a military unit. regimen is a prescribed course of medical treatment

10. L126: “ex.” –nonstandard abbreviation, should be e.g.,

11. L144: “factorwere” – separate conjoined words

Research limitations/concerns:

1. L221: Limitations should also include mention that more patients in the standard treatment group were smokers, as smoking = worse prognosis

2. Needs contextualization: Pain dropped from 3.5 to 3.1 in the treatment group and 5.2 to 5.0 in the standard care group. Given that these are derived from a 0-10 scale, I doubt that these are of clinical importance.

3. Needs clarification: “318 patients were screened for inclusion into the functional medicine group. … The main reasons for exclusion from the study were … escalation of medical therapy for PsA or RA.” If the treatment did not work and drugs were escalated, then the patients were excluded from the final calculations, thereby biasing for treatment efficacy? 264 eliminated from 318 = 83% eliminated.

4. Zero mention of cost, therefore impossible to consider cost-effectiveness. However, given the notoriously high cost of functional medicine care, this is an important omission.

Reviewer #2: The authors report the benefits of functional medicine, an adjuvant form of therapy, in patients with chronic inflammatory arthritis, specifically rheumatoid arthritis and psoriatic arthritis. The study is original and interesting and opens a new avenue for an adjuvant approach to this type of disease.

Before considering publication of this manuscript, this reviewer has the following questions and suggestions.

1. Please define the abbreviation PROMIS in the abstract

2. In the introduction, delve deeper into the concept of functional medicine and what the benefits of this form of treatment are based on. For example, substantiate how this model can influence the immunoinflammatory pathways of both diseases, or if there is any study that has analyzed this intervention on intestinal dysbiosis found in both diseases.

3. Although the authors provide a good methodological description, fundamental information is missing. For example, we know nothing of the pharmacological regimens of both groups; Nor are we told if both groups were adequately matched for these treatments, including their duration; nothing is said about what the inclusion criteria are for subjecting patients to functional medicine. In sum, the authors must provide detailed information on both groups, including information on both types of arthritis patients.

4. The authors use concepts without a clear definition of them. For example, alterations in intestinal permeability are discussed but it is not said how this alteration was detected. Terminology, such as biotransformation or liver detoxification strategies, is also used, without a clear definition of what this means. Note that many readers are unfamiliar with this terminology and what its measurement standards are.

5. In the sense of the previous paragraph, you mention that the potential overexposure to heavy metals was assessed, but we are not told by what method and what values were used for this purpose.

6. Many readers will not be familiar with the PROMIS tool and its standards. Provide more information about it.

7. Some of the statistical tests used as the ANOVA are based on assuming normality in the distribution of the quantitative variables. In general, this test seems more appropriate to compare the means of 3 or more normal variables. The hypothesis of normality in the distribution of the study variables was tested?

8. To define how relevant this type of intervention can be, it would be useful to provide an effect size of the differences found, not only if these differences were significant or not. Only in this way can we know the magnitude of this intervention on the general health of these patients.

General comment:

As the authors point out, the main and most serious problem of the study is its retrospective design, which generates many uncertainties regarding the reliability of the results. On the other hand, the number of patients is small, and the authors chose two very different types of patients. It is known that, for example, metabolic syndrome, a factor that has a very negative impact on the health of these patients, is much more common in psoriatic than in rheumatoid arthritis, therefore, both entities must be approached differently. In fact, there are already experiences in the literature on the benefits of weight loss on the outcomes reported in PsA. Therefore, it is advisable that the authors undertake a prospective randomized clinical trial to give a clear role to functional medicine in the management of these diseases.

Reviewer #3: The authors conduct a 12-week, retrospective study to investigate the impact of functional medicine on patient reported outcomes in patients with inflammatory arthritis. The data were collected at baseline and 12 weeks based on 54 subjects. The results showed that the reduction in pain, change in PROMIS physical health score and changes in global mental health scores were significantly different between the functional medicine group and the standard of care.

1. Line 102. It’s unclear how the study subjects were recruited. Sampling approach? Recruitment period? It lacks of details.

2. Line 144. Typo, “rheumatoid factorwere summarized…”. Should be “factor were”?

3. Line 145. “to test difference in continuous variables”. It’s unclear what difference are discussing here. Need to be more specific.

4. Line 163. “54 patients seen in the reheumatology department alone during the same time period….” It’s unclear which group it refers to. Based on the numbers, it seems refer to the group with functional medicine; however, if so, how the group receiving the standard of care alone was recruited?

5. Table 1. Two of three primary factors (physical health, pain score) were significantly different at baseline between two groups. Was the type of assigned care determined based on those factors during the recruitment?

6. PLOS authors have the option to publish the peer review history of their article (what does this mean?). If published, this will include your full peer review and any attached files.

Reviewer #1: **Yes: **Alex Vasquez

Reviewer #2: No

Reviewer #3: No

---

## [Author Response · Author response to Decision Letter 0]

25 Aug 2020

Please see responses to all reviewers in the uploaded file entitled Response to Reviewers. Thank you.

---

## [Decision Letter · Decision Letter 1]

28 Sep 2020

The Impact of Functional Medicine on Patient-Reported Outcomes in Inflammatory Arthritis: A Retrospective Study

PONE-D-20-14349R1

Dear Dr. Nicole Droz

We’re pleased to inform you that your manuscript has been judged scientifically suitable for publication and will be formally accepted for publication once it meets all outstanding technical requirements.

Kind regards,

Mahmoud Abu-Shakra, MD

Academic Editor

PLOS ONE

Reviewers' comments:

Reviewer's Responses to Questions

**Comments to the Author**

1. If the authors have adequately addressed your comments raised in a previous round of review and you feel that this manuscript is now acceptable for publication, you may indicate that here to bypass the “Comments to the Author” section, enter your conflict of interest statement in the “Confidential to Editor” section, and submit your "Accept" recommendation.

Reviewer #2: All comments have been addressed

Reviewer #3: All comments have been addressed

2. Is the manuscript technically sound, and do the data support the conclusions?

Reviewer #2: Yes

Reviewer #3: (No Response)

3. Has the statistical analysis been performed appropriately and rigorously? 

Reviewer #2: Yes

Reviewer #3: (No Response)

4. Have the authors made all data underlying the findings in their manuscript fully available?

Reviewer #2: Yes

Reviewer #3: (No Response)

5. Is the manuscript presented in an intelligible fashion and written in standard English?

Reviewer #2: Yes

Reviewer #3: (No Response)

6. Review Comments to the Author

Reviewer #2: Many thanks to the authors for adequately responding to all the queries raised by this reviewer. The manuscript now looks significantly improved.

Reviewer #3: (No Response)

7. PLOS authors have the option to publish the peer review history of their article (what does this mean?). If published, this will include your full peer review and any attached files.

Reviewer #2: No

Reviewer #3: No

---

## [Editor Report · Acceptance letter]

30 Sep 2020

PONE-D-20-14349R1 

The impact of functional medicine on patient-reported outcomes in inflammatory arthritis: a retrospective study 

Dear Dr. Droz:

I'm pleased to inform you that your manuscript has been deemed suitable for publication in PLOS ONE. Congratulations! Your manuscript is now with our production department. 

Kind regards, 

on behalf of

Dr. Mahmoud Abu-Shakra 

Academic Editor

PLOS ONE